# Mining Rock Wastes for Water Treatment: Potential Reuse of Fe- and Mn-Rich Materials for Arsenic Removal

**Barbara Casentini** [1],*, **Marco Lazzazzara** [1], **Stefano Amalfitano** [1], **Rosamaria Salvatori** [2], **Daniela Guglietta** [3], **Daniele Passeri** [3], **Girolamo Belardi** [3] **and Francesca Trapasso** [3]

[1] Water Research Institute, National Research Council of Italy (IRSA-CNR), Via Salaria km 29.300, Monterotondo, 00015 Rome, Italy

[2] Institute of Atmospheric Pollution, Italian National Research Council (IIA-CNR), Via Salaria km 29.300, Monterotondo, 00015 Rome, Italy

[3] Institute of Environmental Geology and Geoengineering, Italian National Research Council (IGAG-CNR), Via Salaria km 29.300, Monterotondo, 00015 Rome, Italy

* Correspondence: casentini@irsa.cnr.it

**Abstract:** The worldwide mining industry produces millions of tons of rock wastes, raising a considerable burden for managing both economic and environmental issues. The possible reuse of Fe/Mn-rich materials for arsenic removal in water filtration units, along with rock properties, was evaluated. By characterizing and testing 47 samples collected from the Joda West Iron and Manganese Mine in India, we found As removal up to 50.1% at 1 mg/L initial As concentration, with a corresponding adsorption capacity of 0.01–0.46 mgAs/g mining waste. The As removal potential was strictly related to spectral, mineralogical, and elemental composition of rock wastes. Unlike rock crystallinity due to quartz and muscovite, the presence of hematite, goethite, and kaolinite, in association with the amorphous fractions of Fe and Al, enhanced the As adsorption. The natural content of arsenic indicated itself the presence of active sorptive sites. The co-occurrence of site-specific competitors (i.e., phosphate) represented a consequent limitation, whereas the content of Ce, Cu, La, and Pb contributed positively to the As adsorption. Finally, we proposed a simplified multiple linear model as predictive tool to select promising rock wastes suitable for As removal by water filtration in similar mining environments: As predicted = 0.241 + 0.00929[As] + 0.000424[La] + 0.000139[Pb] − 0.00022[P].

**Keywords:** mining wastes; iron and manganese minerals; water filtration; arsenic adsorption

## 1. Introduction

Millions of tons of waste rock, overburden, and beneficiation wastes are produced by the global mining industry. Due to their limited economic value and the remote location of most mining settings, over 95% of these materials are disposed of, forming enormous stockpiles in the mining area [1–4]. In mining companies, the cost of waste handling and storage can represents a financial loss around 1.5–3.5% of total costs [5]. The transformation of mining wastes is promoted to pursue a zero-waste circular model economy by evaluating solutions for their re-use [6]. The chemical composition and geotechnical properties of the source rock determine which uses are most appropriate and whether reuse is economically feasible. Possible second life pathways of solid mining wastes include the recovery of critical raw materials, the use as backfill materials for open voids, the extraction of valuable minerals and metals from low-grade resources, their application as landscaping materials and capping materials for waste repositories, substrates for mine revegetation, and civil engineering

constructions [1,2,7]. Among mining materials, Fe-, Mn- and Al-rich rock wastes could be recovered as end of life products and converted into adsorbents for water treatment.

In recent years, a range of inexpensive water clean-up technologies have been developed to address the major problem of arsenic contamination in water sources. The adsorption onto filtration units filled with Fe, Mn, and Al (hydr-)oxides phases represents the prominent technological treatment [8–12]. Surface complexation accounts for the high selectivity of the adsorption of arsenic onto iron, aluminum, and manganese (hydr-)oxides [13–16]. Close to point of zero charge, arsenate adsorption through anion exchange could also occur [17]. Iron hydroxide is usually considered to be a superior arsenic adsorbent when compared to aluminum and manganese (hydr-)oxides, due to its highest efficiency at natural pH range [8,10,18]. A large body of the literature is focused on As adsorption studies based on synthetic minerals, such as hematite [19–21], magnetite [19–22], goethite [12,21,22], activated alumina [23,24], gibbsite [16], kaolinite and other clays [25,26], zeolites, and modified zeolites [27–30]. Arsenic adsorption up to 50 mg/g adsorbent were reported, with enhanced adsorption capacity, relying on the homogeneity and activity of adsorption sites [8]. Naturally occurring minerals are more attractive for arsenic water treatment due to their large availability and cost effectiveness. Unlike synthetic iron minerals, the naturally occurring iron ores contain a variety of mineral phases and other elements. Hence, final As adsorption is expected to be lower due to the reduced number of available and accessible sorption sites and interfering and competing ions.

Nevertheless, the need for effective, robust, and low-cost devices for widespread small-scale application (i.e., at the scale of an individual household) increased the interest in testing low-cost waste materials as arsenic adsorbents [8]. Even if their adsorption capacity can be a few mgAs/g, their performance to treat As-rich waters could be satisfactory, especially if applied to drinking water treatment targeted to groundwaters with As concentration below 200 μg/L. Nguyen et al. [31] used a purified and enriched magnetite waste from iron ore mine to treat arsenic-rich waters. This material showed arsenic maximum adsorption capacity of 0.74 mg/g. Zhang et al. [32] tested waste rock from natural iron ores, with hematite and goethite as prevailing mineralogical phases, and maximum adsorption capacity by Langmuir was estimated to be 0.4 mgAs/g. A low-cost material (76% pyrolusite with <10% goethite and quartz) from ferruginous manganese ore efficiently removed As at pH 2–8 from six groundwaters with As concentration in the range 40–180 μg/L [33]. Different tools for the characterization of mining wastes are based on either conventional methods, such as X-ray diffraction and scanning electron microscopy, or advanced approaches, such as synchrotron-based microanalysis and automated mineralogy [34].

Previous studies on mining waste reuse for arsenic removal were based on a limited number of samples with homogeneous mineral distribution. However, since rock wastes in mining stockpiles are highly heterogeneous in terms of mineralogical and chemical composition, the correct identification and selection of suitable materials for the re-use in water treatments will require cross-disciplinary approaches, primarily based on field measurements and sampling site selection.

In this study, we explored the suitability of various mining rock wastes to realize water filters for As removal from contaminated waters. More specifically, we aimed (i) to evaluate if spectral information based on field measurements could help in discriminating materials with different As adsorption potential, (ii) to assess how and to what extent the mineralogical composition and element content of rock wastes can contribute to As removal processes, (iii) to elaborate a pre-screening statistical procedure to identify and select promising materials to be potentially reused in water reclamation practices.

## 2. Materials and Methods

### 2.1. Study Area and Sampling

Joda West Iron and Manganese Mine (JWIMM) is located at about 20 km from Barbil town in Keonjhar, Odisha district, Eastern India (Figure 1). The iron ores belong to the Iron Ore Group

(IOG) and manganese ore deposits. They are confined to shale formation of the Precambrian IOG. In particular, manganese ore bodies are associated with shales, laterite, chert, and quartzite of the IOG and are distributed within the horseshoe-shaped synclinorium, plunging towards NNE over folded towards SW. The shale formation occurs as a core of the synclinorium along Jamda-Koira valley overlying the banded iron formation (See geological details in Supplementary Material, Figure S1). From 1933 onward, the mining lease was granted in favor of Tata Steel. An enormous amount of solid waste is produced each year by mining activities. Valuable material possibly interesting for reuse or recovery, or rock waste not suitable for steel production, is all disposed of in stockpiles. Local workers accumulate wastes into stock deposits as big as mountains (see details in Figure S2), following the color/weight identification of rocks.

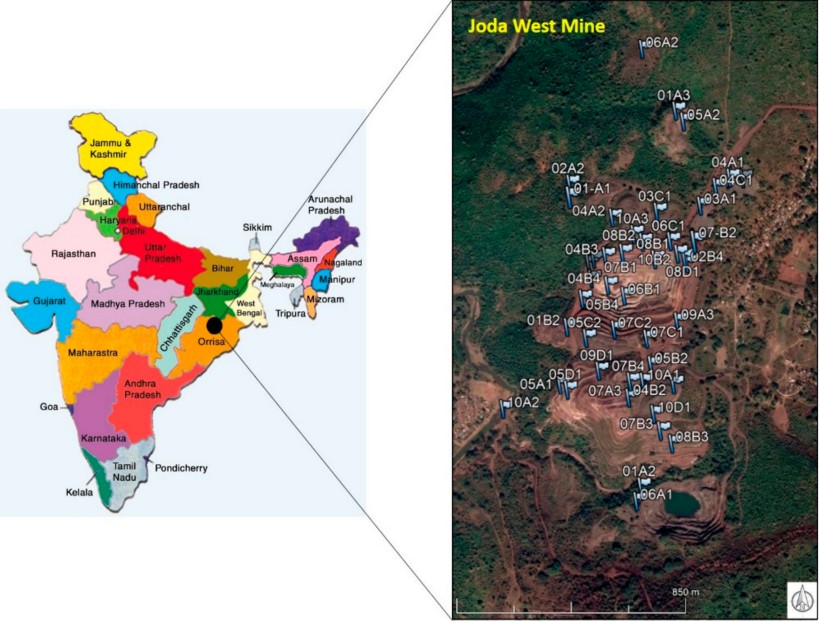

**Figure 1.** Study area with location and codes of sampled rock wastes at Joda West Iron and Manganese Mine.

During November 2018, a field sampling campaign at the JWIMM was conducted by National Research Council (CNR, Rome, Italy) in collaboration with Tata Steel and National Environmental Engineering Research Institute (NEERI, Nagpur, India). In particular, 47 waste rocks were collected from stock deposits and dumps (Figure 1). They were taken in different areas of the mine in order to ensure the heterogeneity of waste materials to be further tested for arsenic removal. Collected samples were micronized under 70 μm by vibrating rotary cup mill at 900 rpm motor speed for further tests.

## 2.2. Rock Waste Characterization

### 2.2.1. X-Ray Diffraction (XRD)

Mineralogical characterization was carried out by XRD analysis on micronized samples with a fully automated AXS D8 Advance diffractometer (Bruker, Billerica, MA, USA) operating in reflection mode with θ-θ geometry, equipped with high-resolution energy dispersive 1-D Lynxeye XE detector opening 3° in 2θ. Measurement parameters used were CuKα, 40 kV, 30 mA, 2.5° Soller collimators, 0.6 mm divergence slit, anti-scatter screen, scan angle (2θ) = 0–70°; step width (2θ) = 0.02°; counting time 0.3 sec per step. Diffraction data were elaborated with DIFFRAC.EVA software and identified using Crystallography Open-Access Database (COD, www.crystallography.net/cod/).

After fitting the major peaks, a semi-quantitative analysis was performed based on the XRD peak relative heights and reference intensity ratio (RIR) values. RIR is the ratio between the intensities of

the strongest line of the compound of interest and the strongest line of corundum in a 1:1 mixture (by weight). The quality of the results depends on the graphic adjustment of the *y*-scale values of each XRD peak. Moreover, the method assumes that the peak height is proportional to the net area of the peak, which may be different for different minerals. An approximate crystallinity index was given as maximum counts in XRD spectra. In addition, the percentage of amorphous phases, as semi-quantitative indicator, was calculated for Al, Mn, and Fe, major adsorption phases, as %Amorphous = %TotalXRF − %CrystallineXRD. The crystalline contribution was calculated based on the sum of all percentage of element contained in each mineral form containing it (Supplementary Materials, Table S1).

### 2.2.2. X-Ray Fluorescence (XRF)

The chemical composition of samples was assessed by an X-ray EDS fluorescence analysis carried out by XEPOS HE spectrometer (AMETEK, Berwin, PA, USA), optimized for heavy elements with max power of 50 W and max voltage of 50 kV. The calibration curves were constructed using certified materials (OREAS, https://www.ore.com.au/) and the common linear model developed by Lucas-Tooth and Price [35].

### 2.2.3. Spectral Characterization

The spectral signature of each undisturbed rock waste sample was recorded using a field hyperspectral spectrometer (FieldSpec FR3 PRO, Analytical Spectral Devices-ASD, Boulder, CO, USA) operating in visible, near-infrared (NIR) and short-wave infrared (SWIR) domain (0.35–2.5 μm). We intentionally selected only the reflectance values related to red band resampled according to band 4 in Sentinel image (range 0.645–0.683 μm), since it was more representative of the target mineral phases (i.e., Fe minerals). A white Spectralon®panel (regarded as a Lambertian reflector) was used as reference to calculate the reflectance of the sample, expressed as ratio to reference (unitless).

### 2.3. Batch Tests for Arsenic Removal

Batch tests were carried out to evaluate arsenic removal capacity of all the 47 sampled rock wastes. Arsenic(V) stock solution (1000 mg/L) was prepared using $Na_2HAsO_4 \cdot 7H_2O$ (Fluka). Standards in the range 1–100 μg/L were prepared by dilution. To test arsenic adsorption properties of samples mining waste, 20 mL Milli-Q spiked to initial concentration of 1000 μg/L As(V) was placed in contact with 20 ± 2 mg of sample (liquid/solid ratio of 1 g/L). Solution pH was 7.0 ± 0.5. Arsenic adsorption capacity was expressed as mg of adsorbed As per grams of mining waste (mgAs/g). Initial As concentration of 1000 μg/L was selected to keep concentration sufficiently high and not far from arsenic levels typically found in groundwaters (20–200 μg/L). Samples were mildly shaken onto orbital shaker at 160 oscillations/min for 5 h. Samples were filtered on 0.2 μm acetate cellulose filters. Arsenic in solution was measured, following appropriate dilution, by Atomic Absorption Spectrometry AAnalyst 800 (Perkin Elmer, MA, USA) equipped with Ir-coated THGA furnace (range of linearity 0–100 μg/L). Duplicate samples were carried out on 10% of entire dataset. According to As adsorption capacity, mining rock wastes were classified into two groups (i.e., "not suitable (-)", "suitable (+)") by using the median as discriminatory value (i.e., 0.249 mgAs/g). Samples with significantly higher removal efficiency (i.e., 0.35–0.5 mgAs/g) were classified as "promising (++)".

### 2.4. Statistical Analysis

Descriptive statistic and multivariate analyses were performed on the dataset using the freeware software PAST [36]. Non-parametric statistics were applied because the normal distribution was rejected for many of the measured variables. The Kruskal–Wallis test was used to verify the equality of medians per single variable between the three groups of sampled materials with different As adsorption potential. Spearman's correlation coefficients ($r_s$) were calculated between all pairs of variables.

When required, the min-max data normalization was applied $y = (x - min)/(max - min)$, where min and max are the minimum and maximum values of selected parameter.

Considering the multivariate dataset, the significance difference between sample groups was tested by the non-parametric multivariate analysis of variance (PERMANOVA), based on the Euclidean distance measure of normalized data. The similarity percentage analysis (SIMPER), based on the Bray–Curtis dissimilarity matrix, was used to calculate the percentage contribution of each variable (i.e., among the mineralogical phases, the major and trace elements) to the overall dissimilarity between the sample groups [37].

The factor analysis (CABFAC) was used to verify whether all information conveyed by the analyzed variables ($n = 53$) could be used to consistently predict the As adsorption potential in comparison to the measured As adsorption values [38]. Data were also modeled using multiple linear forward stepwise regression through SigmaPlot (v. 11.0)(Dundas Software LTD, GmbH, Germany). The goodness of fit was then evaluated in terms of coefficient of determination ($R^2$) and root mean square error (RMSE), where $R^2$ represents the relative measure of fit (trend prediction), while RMSE is an absolute measure of fit (model accuracy).

Coefficient of Determination

$$R^2 = \frac{\sum(x_m - \overline{x_e})^2}{\sum(x_m - \overline{x_e})^2 + \sum(x_m - x_e)^2} \tag{1}$$

Root Mean Square Error

$$RMSE = \sqrt{\frac{1}{n}\sum_{i=1}^{i=n}(x_m - x_e)^2} \tag{2}$$

where $x_m$ is the value given by the model, $x_e$ is the experimental data, and $\overline{x_e}$ is the mean value of experimental dataset. To have an estimate of the overall prediction error, the value of RMSE was then divided by mean of predicted values.

Predicted data were divided into correctly assigned (high and low) and mistakenly classified (false high and false low).

## 3. Results

### 3.1. Mineralogical and Chemical Composition of Rock Wastes

The entire samples dataset is shown in Table S2a–c, while a summary is reported in Table 1a,b. Among mineralogical phases, hematite, goethite, kaolinite, pyrolusite, and quartz were the most frequently found (>30% of sampled materials). Hematite was the dominant Fe mineral with samples showing more than 80% content. Goethite contribution was on average 11.2%. Samples with high crystallinity (i.e., 4000 counts in XRD spectra with sharp high peaks), were characterized by the presence of quartz (54.3–87.8%). Radiometer signal (in red band) ranges from 0.07 (dark minerals rich in Mn or hematite) to 0.38 (higher reflectance characterized by whitish minerals, like quartz, muscovite, or kaolinite). Mining rock wastes showed a heterogeneous composition with Fe, Mn, and Al as main constituents (51.3%, 14.3%, and 6% average content, respectively), with rocks having Fe and Mn above 70%. Arsenic was also naturally present in selected samples in the range 2.8–139.8 mg/kg, with a mean value of 36.2 mg/kg. Phosphorus concentration was one order of magnitude higher than As (mean = 350.1 mg/kg), due to the presence of phosphate minerals, such as berlinite, zanazziite and hopeite (Table S2). Sulfide concentration was low (mean = 140.9 mg/kg).

**Table 1.** Major properties of sampled rock wastes. Mineralogical properties (**a**), including major mineral phases (%) occurring in >30% of total samples are shown along with the content of major (**b**) and trace (**c**) elements.

| Mineral Properties | Red Band Reflectance | Crystallinity Index | Hematite ($\alpha$-Fe$_2$O$_3$) | Goethite ($\alpha$-FeOOH) | Kaolinite (Al$_2$Si$_2$O$_5$(OH)$_4$) | Pyrolusite (MnO$_2$) | Quartz (SiO$_2$) |
|---|---|---|---|---|---|---|---|
| Mean | 0.14 | 3015 | 37.4 | 11.2 | 10.7 | 4.5 | 19.0 |
| Median | 0.13 | 2000 | 41.4 | 8.0 | 0.0 | 0.0 | 10.7 |
| Std Dev. | 0.07 | 3219 | 20.1 | 9.7 | 15.3 | 10.4 | 23.2 |
| Min | 0.07 | 800 | 0.0 | 0.0 | 0.0 | 0.0 | 0.0 |
| Max | 0.38 | 13000 | 81.5 | 34.1 | 53.4 | 57.6 | 87.8 |

(**a**)

| Major Elements (%) | Al | Fe | Mn | Ca | K | Mg | Si | Ti |
|---|---|---|---|---|---|---|---|---|
| Mean | 6.0 | 51.3 | 14.3 | 0.10 | 1.1 | 0.15 | 9.7 | 0.42 |
| Median | 5.3 | 54.8 | 5.1 | 0.10 | 0.7 | 0.07 | 6.4 | 0.34 |
| Std Dev. | 3.5 | 22.2 | 18.1 | 0.05 | 1.2 | 0.28 | 9.6 | 0.29 |
| Min | 0.5 | 8.9 | 0.4 | 0.02 | 0.1 | 0.01 | 0.8 | 0.04 |
| Max | 17.2 | 87.4 | 75.8 | 0.24 | 5.5 | 1.39 | 40.5 | 1.33 |

(**b**)

| Minor Elements (mg/kg) | As | Ce | Cr | Cu | La | Mo | Ni | P | Pb | Rb | S | Y | Zn |
|---|---|---|---|---|---|---|---|---|---|---|---|---|---|
| Mean | 36.2 | 49.6 | 290.5 | 28.5 | 46.6 | 65.8 | 221.2 | 350.1 | 272.7 | 42.8 | 140.9 | 44.3 | 162.8 |
| Median | 32.7 | 38.0 | 241.9 | 24.6 | 34.1 | 33.1 | 171.0 | 350.7 | 231.0 | 41.8 | 109.8 | 32.8 | 150.4 |
| Std Dev. | 34.7 | 41.5 | 191.3 | 14.9 | 51.9 | 65.1 | 293.2 | 114.5 | 260.1 | 14.2 | 101.5 | 38.6 | 90.9 |
| Min | 2.8 | 1.5 | 52.6 | 8.1 | 1.5 | 0.2 | 0.8 | 93.3 | 0.3 | 18.4 | 26.6 | 1.1 | 29.4 |
| Max | 139.8 | 175.9 | 846.2 | 82.4 | 284.2 | 263.0 | 1914 | 678.8 | 1360 | 112.6 | 473.5 | 192.7 | 432.6 |

(**c**)

## 3.2. Arsenic Removal Capacity

Arsenic removal ranged from 1.2% to 50.1%, with a calculated adsorption capacity of 0.01–0.456 mgAs/g. Variation in duplicate samples were in the range 5.4–14.3%. Adsorption capacity distribution showed a mean value of 0.255 and median of 0.249 mgAs/g. Sampled rock wastes with a potential As adsorption lower than the median were classified as "not suitable" (45% of total samples), while 55% of samples are classified suitable, including 13% of them showing promising capacity for As removal application (Figure 2).

## 3.3. Influence of Spectral, Chemical, and Mineralogical Parameters on As Adsorption

The three identified groups were significantly different in terms of As adsorption and reflectance (Kruskal–Wallis test, $p < 0.05$). Reflectance values above 0.2 were only found in the group with lower As adsorption capacity (Figure 2). Samples with >0.2 reflectance in red band were characterized by the presence of muscovite (13.7–35.1%), and quartz (35.6–78%).

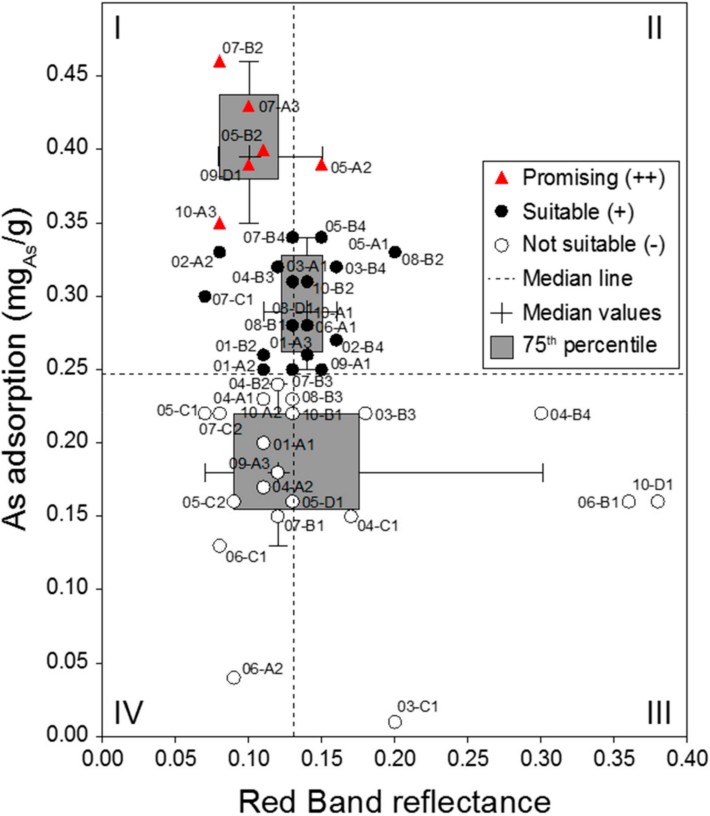

**Figure 2.** Bi-directional box plots of Red Band reflectance and As adsorption (mgAs/g) of sampled rock wastes, coded as in Figure 1. The overall median values of the two parameters (dashed lines) were used to discriminate samples with relatively lower and higher transmittance and adsorption potential (i.e., not suitable (-), suitable (+), and promising (++) for As removal). Samples were divided into four quadrants according to their characteristics. Quadrant I: high As adsorption and low Red Band reflectance (26% of samples); Quadrant II: High As adsorption and high Red Band reflectance (30% of samples): Quadrant III: low As adsorption and high Red Band reflectance (19% of samples) and Quadrant IV: low As adsorption and low Red Band reflectance (26% of samples).

The rock waste groups were also proven to be significantly different considering the entire normalized dataset of spectral, mineralogical, and chemical parameters (PERMANOVA, *p* = 0.011).

Samples characterized by lower As adsorption and classified as "not suitable (-)", showed a relatively lower content of hematite, goethite, and kaolinite, along with the prevalence of quartz. Samples 03A1 and 06A2 showed extremely low adsorption capacity (<0.05 mgAs/g) and crystalline hematite above 60% (Figure 3). On the contrary, the concomitant presence of iron minerals, with high content of kaolinite and low contribution of quartz led to the relatively higher arsenic adsorption measured for samples classified as "suitable (+)" and "promising (++)".

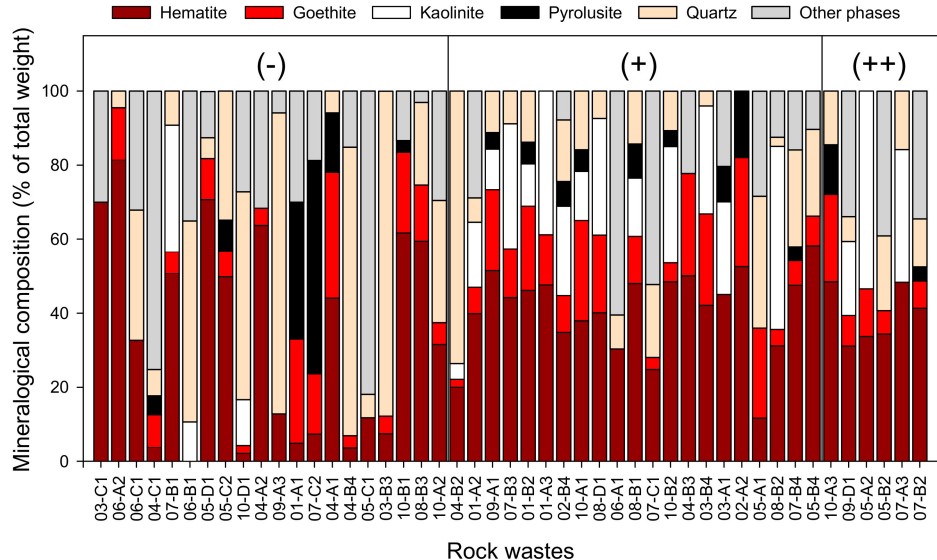

**Figure 3.** Mineralogical composition of sampled rock wastes (i.e., not suitable (-), suitable (+), and promising (++) for As removal), ordered by increasing As adsorption potential (from left to right). Mineralogical phases occurring in less than 30% of the samples were grouped and plotted as "other phases".

The SIMPER tests, carried out on mineralogical phases, major, and trace elements (Table 3a–c), showed that the mineral phases kaolinite, goethite, quartz, and hematite explained most of the overall dissimilarity in the dataset (>30% of cumulative contribution). Notably, the less represented mineral phases with contribution close to 5% were silico-aluminates (staurolite and clinochlore) and phosphate rocks (zanazziite). Among major elements, including % of amorphous (am-) and crystalline (cryst-), Fe, and Al, especially the forms crystalline/amorphous Al and amorphous Fe, were higher in suitable and promising groups, indicating their predominant role in As adsorption (Figure S3). Moreover, the presence of Mn and Si did not promote or inhibit As adsorption.

**Table 2.** Outputs of the similarity percentage analysis (SIMPER) test performed on normalized data on the three sample groups with different As adsorption potential, classified as "not suitable (-)", "suitable (+)", and "promising (++)". The mineralogical phases (**a**), major (**b**) and trace (**c**) elements were tested separately. The parameters were sorted in descending order of percentage contribution (Contrib %) to the observed difference between sample groups. Mean values for each variable and sample group are also reported.

| Mineral Phases % of Total Weight | Contrib % | Mean Values | | |
|---|---|---|---|---|
| | | (-) | (+) | (++) |
| Kaolinite | 10.1 | 2.9 | 16.6 | 18.2 |
| Goethite | 8.4 | 8.8 | 14.2 | 9.7 |
| Quartz | 7.3 | 28.3 | 11.3 | 11.7 |
| Muscovite | 6.3 | 5.9 | 2.1 | 0.0 |
| Hematite | 5.9 | 32.8 | 41.6 | 39.6 |
| Zanazziite | 5.5 | 1.1 | 0.5 | 0.0 |
| Staurolite | 4.7 | 4.3 | 1.6 | 0.0 |
| Clinochlore | 4.7 | 2.9 | 1.1 | 0.0 |
| Gjerdingenite-Fe | 4.4 | 1.6 | 0.8 | 0.0 |
| Gibbsite | 3.7 | 0.8 | 4.1 | 7.9 |
| Birnessite | 3.6 | 1.4 | 0.9 | 1.3 |
| Krettnichite | 3.5 | 0.0 | 0.2 | 1.6 |
| Ellenbergerite | 3.2 | 0.0 | 0.0 | 3.4 |
| Ferrierite-Na | 3.2 | 0.0 | 0.0 | 2.5 |
| Hopeite | 3.2 | 0.0 | 0.0 | 1.3 |
| Pyrolusite | 3.1 | 6.1 | 3.4 | 2.9 |
| Siderite | 2.8 | 0.0 | 0.6 | 0.0 |
| Gehlenite | 2.1 | 0.0 | 0.2 | 0.0 |
| Inesite | 2.1 | 0.0 | 0.4 | 0.0 |
| Kogarkoite | 2.1 | 0.0 | 0.4 | 0.0 |
| Berlinite | 2.0 | 0.3 | 0.0 | 0.0 |
| Chalcophanite | 2.0 | 0.2 | 0.0 | 0.0 |
| Lazurite | 2.0 | 0.8 | 0.0 | 0.0 |
| Magnesiochromite | 2.0 | 0.3 | 0.0 | 0.0 |
| Pyroxene-ideal | 2.0 | 1.3 | 0.0 | 0.0 |

(**a**)

| Major Elements % | Contrib % | (-) | (+) | (++) |
|---|---|---|---|---|
| cryst-Al | 10.8 | 2.4 | 4.2 | 5.7 |
| am-Fe | 10.2 | 17.4 | 17.7 | 18.6 |
| am-Al | 9.7 | 2.8 | 2.4 | 1.3 |
| Fe | 9.4 | 46.6 | 56.0 | 52.4 |
| Mn | 6.8 | 17.8 | 10.3 | 15.2 |
| Si | 6.8 | 11.8 | 8.3 | 7.0 |
| cryst-Fe | 6.8 | 21.8 | 15.2 | 21.0 |
| K | 6.7 | 1.6 | 0.8 | 0.6 |
| am-Mn | 6.7 | 13.4 | 7.7 | 12.4 |
| Ca | 6.3 | 0.1 | 0.1 | 0.1 |
| Ti | 5.9 | 0.4 | 0.4 | 0.4 |
| Al | 5.4 | 5.1 | 6.5 | 6.9 |
| Mg | 5.0 | 0.2 | 0.1 | 0.1 |
| cryst-Mn | 3.6 | 4.7 | 2.8 | 2.7 |

(**b**)

**Table 3.** *Cont.*

| Minor Elements mg/kg | Contrib % | (-) | (+) | (++) |
|---|---|---|---|---|
| As | 10.7 | 26.1 | 39.5 | 60.6 |
| Ce | 10.5 | 38.4 | 52.8 | 78.1 |
| Mo | 9.7 | 72.5 | 54.0 | 81.9 |
| Cr | 9.2 | 312.0 | 258.0 | 321.0 |
| Zn | 8.8 | 155.0 | 159.0 | 205.0 |
| S | 8.7 | 130.0 | 151.0 | 143.0 |
| Cu | 7.1 | 24.3 | 30.7 | 36.1 |
| P | 6.8 | 394.0 | 305.0 | 347.0 |
| La | 6.8 | 34.8 | 46.8 | 86.8 |
| Y | 6.6 | 54.6 | 41.0 | 19.6 |
| Pb | 6.4 | 187.0 | 291.0 | 510.0 |
| Ni | 5.2 | 187.0 | 175.0 | 494.0 |
| Rb | 3.6 | 43.0 | 42.8 | 41.9 |

(c)

Bivariate correlation plots evidenced that arsenic adsorption (mgAs/g) was significantly correlated with As, Ce, Cu, P, Pb, and Y naturally occurring in the sampled materials (mg/g, XRF measurements). The presence of Ce, Cu, and Pb led to an increase of As adsorption, while the presence of P and Y was inversely correlated. Despite data of As, Pb and Ce corresponding to their LOD value (2.8, 0.3, 1.5 mg/kg, respectively) being close to the *x*-axis (Figure 4), the As adsorption was measurable since the adsorption driving forces in rock waste were dependent on a combination of factors.

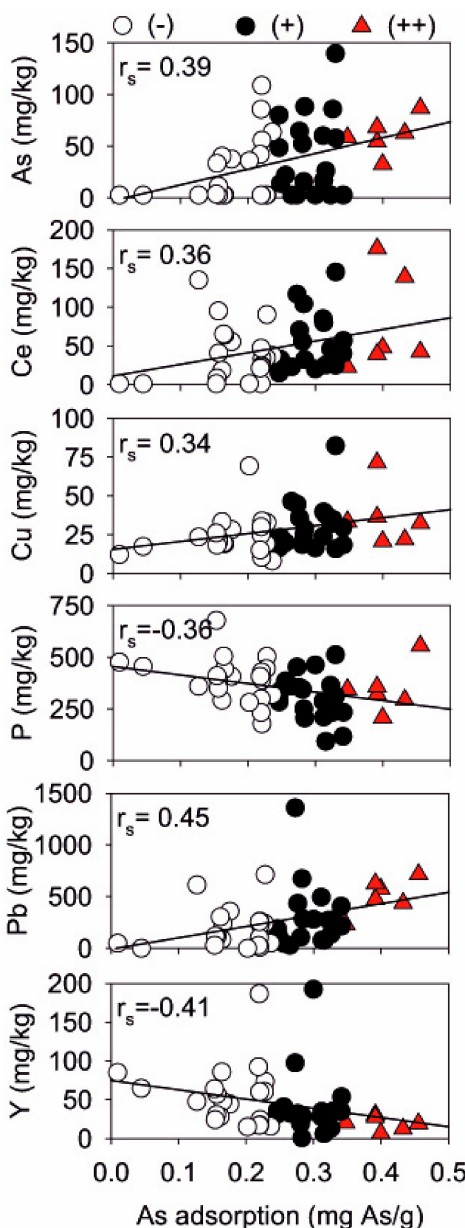

**Figure 4.** Bivariate plots and significant Spearman correlations (r$_s$) between the As adsorption potential and the content of selected trace elements (*p* < 0.02, always). Symbols indicate samples not suitable (-), suitable (+), and promising (++) for As removal.

### 3.4. Predicting Arsenic Removal by Wastes Characteristics of Fe- and Mn-Rich Ores

Factor analysis, based on the entire dataset, was tested to formulate an arsenic adsorption predictive model. Arsenic adsorption could be predicted by all variables, with and R$^2$ = 0.6, RMSE of 0.06 corresponding to 22.8% prediction error. To our scope, the error that more affected materials selection and further filter performance was the one represented by false high, that is those samples predicted as suitable adsorbents that ended up not being suitable. In the case of false low materials, the error represents an underestimation of our materials and leads only to non-inclusion of wastes that are possibly good adsorbents. False high error corresponded to 9%.

Factor analysis proved the possibility of building a predictive model. Thus, we developed a simplified predictive tool by extrapolating a multiple linear model based on forward stepwise regression (Figure 5). The resultant equation is:

$$\text{As predicted} = 0.241 + 0.00929[\text{As}] + 0.000424[\text{La}] + 0.000139[\text{Pb}] - 0.00022[\text{P}] \tag{3}$$

where As predicted is given in mg/g, and chemical concentrations of single elements, measured by XRF, are in mg/kg.

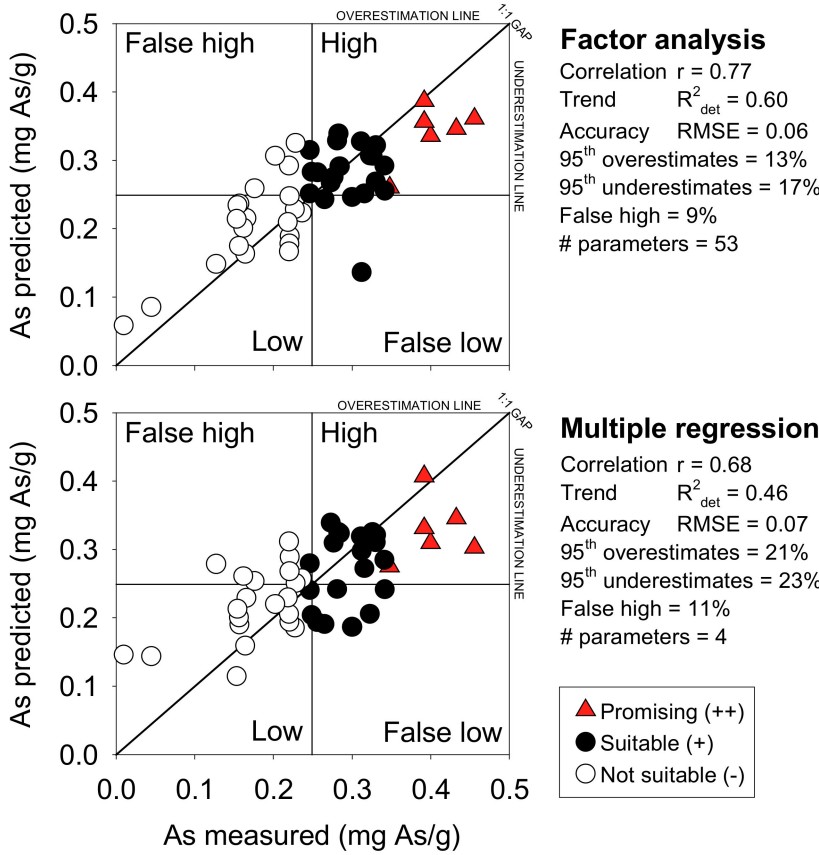

**Figure 5.** Multiparameter linear regression results based on (**a**) factor analysis performed on 53 parameters and (**b**) multiple linear stepwise forward regression, based on four selected parameters (As, La, Pb, and P). Goodness of fit are expressed through $R^2$ and root mean square error (RMSE). Percentiles lines (95%) of prediction are plotted (dashed lines). Symbols indicate samples not suitable (-), suitable (+), and promising (++) for As removal.

Among heavy metals, the one mostly affecting As adsorption ability is Pb, while among rare earth elements, lanthanum contribution was dominant. The presence of P in the materials limited the adsorption potential. The presence of natural arsenic enrichment in the sample was the major predictor for As adsorption potential capacity. The prediction using a multiple linear regression model showed $R^2 = 0.46$, RMSE of 0.067, and error of prediction 26.3%, while false high corresponded to 11%. Overestimation of 21% and underestimation of 23% was calculated using multiple linear model based on 95th percentile interval.

## 4. Discussion

Mining activities, together with construction/demolition and manufacturing, contributed to nearly 74% of all wastes disposed of in the European Union [39]. North America produces more than 10 times as much solid mine waste as municipal solid waste per capita [34]. In India, more than 200 million tons

of non-hazardous inorganic solid wastes are being generated every year, out of which about 80 million tons are mine tailings/ores [40].

At Joda West mine, Fe, Mn and Al (hydr-)oxides, clay mineral, and quartz were the predominant mineral phases of collected rock wastes. The natural arsenic content reached up to 140 mg/kg, (higher than soil baseline concentration generally 5–10 mg/kg). In different mining environments (e.g., gold mines), arsenic content in rocks was reported in the order of g/kg, due to the presence of primary and secondary minerals [41]. Among transition metals, Cr, Ni, Pb, and Zn showed higher concentrations than average values in soils and sediments reported by FOREGS European geochemical Atlas [42] but much lower than their reported maximum values (Table S3).

First exploratory adsorption tests allowed us to test and select materials that could be potentially interesting for the realization of filters for treating As-rich waters. In heterogeneous natural iron oxides, adsorption of 0.3–0.5 mg/g of arsenate were commonly found [12,19,20]. Some of the tested samples showed a satisfactory As adsorption capacity (>0.25 mgAs/g at Ci = 1 mgAs/L). At this concentration, treated magnetite waste from iron ore showed adsorption below 0.2 mg/g [31]. Chakravarty et al. [33] tested a ferruginous manganese ore material, mainly constituted by pyrolusite and goethite, and concentration in condition similar to this study resulted in 0.2 mgAs/g adsorption according to Langmuir As(V) equation. The As adsorption capacity of three hematite-rich iron ore samples was 0.17–0.48 mgAs/g [32]. The presence of quartzite and less reactive clays at high crystallinity (e.g., muscovite) were the major limiting factors for adsorption.

Since the reflectance spectrum of rocks depends on their mineralogical composition [43,44], we noted that the reflectance in red band range, measured by hyperspectral field radiometer on undisturbed rock wastes, provided valuable indications for the on-site pre-selection of materials with lower As removal potential. The rock wastes with a limited availability of adsorption sites could be discarded, with no need to carry out further measurements. A variety of reflectance spectroscopy-based applications, relying on the spectral signatures of minerals able to bind/sorb metal(loids), have been developed to promote indirect detection and avoid expensive laboratory measurements. Pallottino et al. [45] realized a predictive model for As contamination in calcareous soil surrounding thermal springs based on the diffuse VIS-NIR spectral reflectance. In that study, As content was largely associated (>46%) with the sole $CaCO_3$ phase. According to our outcomes, a first pre-screening step could be used to exclude spotted materials containing high amounts of quartzite and muscovite, but the presence of dark-red minerals (amorphous and crystalline) cannot be directly discriminated through the spectral signals in the red band, given the observed mineralogical complexity and heterogeneity of As adsorption phases. Appropriate spectral information should be collected by better refining band selection in order to exclude the less adsorptive materials (i.e., quartz and muscovite) and to identify good adsorbents (i.e., kaolinite).

The presence of iron minerals (goethite and hematite), together with Al-rich kaolinite, contributed the most to As adsorption. At natural pH range of 6–8, the adsorption onto iron (hydr-)oxides is the most competitive, since Fe-based materials have a favorable surface charge (pHpzc 7-9) for oxyanion adsorption, while Mn oxides are mostly negatively charged (pHpzc 2–3). Adsorption of aluminum (hydr-)oxides is known to be maximum at pH 4–5 [10]. On the contrary, Fe-Al binary oxides showed to be attractive adsorbents for both As(V) and As(III) removal from contaminated waters [46]. The ability of Mn dioxides to sorb As(III) and As(V) appeared to be related also to materials with highly ordered pyrolusite having low specific surface (7.9 m$^2$/g). Conversely, poorly crystalline birnessite has higher specific surface area of 27.7 m$^2$/g [47].

The presence of Mn minerals, either amorphous or crystalline, lowered As adsorption. Arsenic adsorption onto hydroxides was correlated to Fe and amorphous phases, characterized by edge structures more efficient in hosting arsenate ions than crystalline minerals. For example, the transformation of amorphous FeOOH to crystalline FeOOH would reduce sorption sites and surface area, thus lowering the number of ions that can be adsorbed [15,41,48]. We found that amorphous fraction of Fe and Al were important in promoting As adsorption. Pedersen et al. [49] observed a

decrease in adsorbed arsenic clearly correlated with the transformation of ferrihydrite and lepidocrocite into more crystalline phases as goethite and hematite. Pigna et al. [50] showed better As removal capacity of non-crystalline Al(OH)x than gibbsite. Fine-grained and poorly crystalline Mn oxides showed good adsorption properties, even if in Mn-ores pyrolusite (most stable and abundant) and birnessite minerals are often encountered [51]. Fe, Al, and Mn minerals with medium grade crystallinity are responsible for As adsorption, since crystallization process kinetic and environmental conditions might induce defects in crystalline structures, which are suitable as adsorption sites.

In line with literature reports, the correlation of As adsorption with the presence of rare earth elements (Ce and La) and transition metals (Pb, Cu) suggested that their variation mostly explained changes in As adsorption capacity [10,52–55]. These elements could be incorporated in mineral structures or adsorbed on specific sites. The adsorption of heavy metals onto clay and oxides surface might cause a pHpzc shift towards higher values, thus rendering surfaces more positive at higher pH and promoting adsorption of oxyanions [8,56–58]. Fe (hydr-)oxides structure may incorporate metals cations and adsorb As-ions more effectively, due to a better matching of ion size and orientation, also by shortening the atom-to-atom distances between adsorbent and adsorbate [59]. Mohapatra et al. [60] modified goethite surface by doping Cu(II), Ni(II), or Co(II) to enhance arsenate uptake capacities. Lu et al. [61] observed that the presence of Pb during the process of ferrihydrite transformation to hematite induced the formation of nanoparticles with a loose and porous structure in comparison with the compact structure of pure hematite nanoparticles.

The presence of bivalent cations (namely, Ni(II), Co(II), Mg(II)) were reported to enhance As adsorption capacity [33]. Natural and modified enriched clays with exchangeable cations and anions have been widely tested as adsorbents for water treatment [62–65]. In pillared or intercalated clays with transition positively charged metals, clay sheets increase each other's distances. Adsorption increased due to change in surface area and charge, with positively charged surface enhancing penetrability of As oxyanions [66]. Na et al. [67] demonstrated that Ti-pillared montmorillonite was an efficient material for the removal of arsenate and arsenite from aqueous solutions. Doušová et al. [68] proved that pre-treatment of low-grade clay materials Fe (Al, Mn) salts can significantly improve their sorption affinity to As oxyanions. A simplified multiple linear model was proposed, based on XRF measurements, as a predictive tool to guide mining wastes selection to realize removal filters for As contaminated water. As predicted = 0.241 + 0.00929[As] + 0.000424[La] + 0.000139[Pb] − 0.00022[P]. This model applicability is site-specific and strictly related to the mineralogical, geological and chemical context encountered at JWIMM. The natural presence of arsenic in the sampled materials was one of the best predictors for As adsorption, thus indicating that rock wastes kept their original and natural As-adsorbing affinity. On the contrary, higher concentrations of Y and Rb were found at low As adsorption levels only. These two elements are found in association with phosphate rocks [69,70]. Due to chemical similarity, phosphate is also known to be a competitor for the As adsorption sites and, together with silicate, a major interfering ion for As removal processes onto oxides [71–74]. Furthermore, waste materials active for arsenate could be also successfully tested for phosphate removal to reduce P-load from surface water and promote its recovery as critical raw material [75,76].

Overall, due to the elevated presence of iron oxides and the co-occurring kaolinite in most suitable samples, the factors that turned out to be more significant to differentiate potential adsorption capacity were elements adsorbed onto major phases, which were able to positively modify surrounding adsorption site structure. The possible release of As together with other potentially toxic metals from reused rock wastes should be consciously investigated in the long term to promote safer applications, especially if intended for human consumption purposes.

## 5. Conclusions

Mining rock wastes, accumulated in stockpiles at Fe-Mn ores, showed a good potential to be reused for water treatment, due to the presence of iron minerals and kaolinite clays. The adsorption capacity of suitable materials was not exceptionally high (0.25–0.46 mgAs/g), but satisfactory for

treating As-rich groundwaters. The use of spectral, mineralogical and chemical information proved useful to select heterogeneous materials and promisingly suitable to remove arsenic from contaminated waters. Given the significant correlation with As adsorption, the role of positively charged ions in the structure of clays and oxides should be considered specifically, also evaluating thermodynamic and kinetic properties which may affect the efficiency of filtration-based As removal by reused mining rock wastes.

**Supplementary Materials:** The following are available online at http://www.mdpi.com/2073-4441/11/9/1897/s1, Figure S1: Geological map of Joda West Mine area, Odisha, India; Figure S2: Joda West mining site and waste stockpiles; Figure S3: Amorphous (dark bars) and crystalline (light bars) contribution to overall presence of Fe (blue), Mn (green) and Al (grey) divided into As adsorption groups: (-) "not suitable", (+) "suitable) and (++) "promising". Samples were ordered by increasing As adsorption capacity (from left to right). Semi-quantitative contribution (%) of amorphous fraction for Fe, Mn and Al is given by subtracting by XRD data (% crystalline phases) from XRF measurements (% Total element).Table S1: Part I: Mineral formula and contribution (%) of Fe, Al, Mn to each phase for the calculation of Crystalline phase used to derive semi-quantitative amorphous value. Part II: Other mineralogical phases; Table S2: (a) All samples codes, site and color description, As removal efficiency (% and mg Arsenic/g material), spectral information and contribution of Crystalline and amorphous fraction of selected major elements (Fe, Al, Mn); (b) All samples codes with major (%) and minor (mg/kg) elements measured by XRF; (c) All samples codes with mineralogical phases major (PartI) and minor (PartII) (%) measured by XRD. Other phase is the sum of all minor phases. n.d. = not detectable phases; Table S3: European soil and sediments major and minor elements content (mg/kg) (FOREGS geochemical Atlas.

**Author Contributions:** Conceptualization: B.C. and S.A.; Data curation: B.C., D.G. and F.T.; Formal analysis, B.C., S.A and G.B.; Investigation: M.L., R.S., D.G., D.P., G.B., and F.T.; Resources: D.G., F.T. and D.P; Supervision:, B.C.; Validation: S.A.; Visualization: B.C. and S.A.; Writing—original draft: B.C.; Writing—review & editing, S.A., R.S. and F.T.

**Funding:** The authors are grateful to TECO Project ICI+/2014/342-817-"Technological Eco - Innovations for the Quality Control and the Decontamination of Polluted Waters and Soils", for partly funding sampling campaign.

**Acknowledgments:** The authors are grateful to the staff of Tata Steel for providing general support during fieldwork.

**Conflicts of Interest:** The authors declare no conflict of interest.

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
