# Peer review of "Mining Rock Wastes for Water Treatment: Potential Reuse of Fe- and Mn-Rich Materials for Arsenic Removal"

_water, doi:10.3390/w11091897_

Round 1

Reviewer 1 Report

The focus of the manuscript is to explore selected material absorption of As that can use for treating As-rich waters in which reuse of Fe/Mn-rich material of mining rocks are investigated for arsenic removal capability.  47 waster rocks were collected from Barbil town in eastern India and micronized under 70 um , which was used to remove the arsenic of standard sample (Na2HAsO4*7H2O, range from 1-100 ug/L), Waste material was characterized using X-Ray diffraction and X-Ray fluorescence.  Hyperspectral spectrometer was utilized for the spectral signature of rocks. Arsenic concentration in the standard was measured using absorption spectrometry.  The samples were filters using 0.2 um acetate cellulose filters. Sample rocks were 45% of not suitable (-) and 55% of suitable (+), including 13% is showing promising capacity. Heavy elements (Pb and lanthanum) showed a dominant contribution of As adsorption.  P showed limited absorption potential. Also, the presence of As plays a crucial role in As absorption.

 The manuscript is organized and well written. The author recommended improving the writing style. The following revision is recommended before publication:

Line 103, Author can switch to “from stock deposits and dumps”  to “from stock deposits and dumps of different locations.” It is hard to understand the section 2.3. “Batch tests for arsenic removal.” Author needs to improve the writing style, especially between Line 145 and 149. Line 151: It is not clear what and why appropriate dilution is necessary. Is it due to out of range of spectrometer? Author needs specify. Line 146: Is it “Milli-Q” instead of “MilliQ”? Line 212-214: Author can rewrite as 55% of samples are classified suitable, including 13% of them showed promising capacity for As removal application. Figure 2: what is the unit of Red Band reflectance? It is not clear why the percentage (26%, 30%, and 19%) inside the Figure. The author needs to describe in the caption.    Line 301: LOD is not defined previously.

Author Response

The authors are grateful to the reviewer for his/her comments and have been implemented them, if necessary, wherever requested. English has been revised throughout the manuscript.

Reviewer 2 Report

Research is interesting and quite comprehensive. However, they require some corrections and additions.

Line 19 - 0.01-0.46 mgAs/g. It is not clear what gram "g" refers to.

Line 55 - 50 mg/g. It is not clear what gram "g" refers to.

Line 63 - There is "dsorbents", and it should be "adsorbents".

Line 144 - Is 1000 mg/l, or shouldn't be 1000 μg/l?

Table 1 - In part "a" no units.

Figure 2 - How to interpret numbers 26%, 26%, 19% and 30% in the corners of the graph?

Lines 264 - 297 are empty.

Lines 298 - 299 - "Bivariate correlation plots evidenced that arsenic adsorption was significantly correlated with As, Ce, Cu, P, Pb, and Y naturally occurring in the sampled materials". Is this a correlation related to the very presence of As, Ce, Cu, P, Pb and Y, or their concentrations.

Table 2 - "Mean values" what does it concern and in which units?

Line 318 - The range of applicability of the model is missing. In my opinion, the authors should extend the discussion of results by analyzing the residuals of obtained multiple regression models. How was the model constructed? Have all measurement series been used or been it verified with separate measurement results?

Sorry. I could not find Fig. S1, S2 and S3 and tables S1, S2 and S3.

Author Response

(The authors gave the same response as above.)
